# Spotlight on P2X7 Receptor PET Imaging: A Bright Target or a Failing Star?

**DOI:** 10.3390/ijms24021374

**Published:** 2023-01-10

**Authors:** Stephan Schmidt, Andreas Isaak, Anna Junker

**Affiliations:** European Institute for Molecular Imaging, University of Münster, Waldeyerstr. 15, 48149 Münster, Germany

**Keywords:** purinergic P2X7 receptor (P2X7R), P2X, antagonists, glioma, CNS, PNS, neuroinflammation, PET tracer, imaging

## Abstract

The homotrimeric P2X7 receptor (P2X7R) is expressed by virtually all cells of the innate and adaptive immune system and plays a crucial role in various pathophysiological processes such as autoimmune and neurodegenerative diseases, inflammation, neuropathic pain and cancer. Consequently, the P2X7R is considered a promising target for therapy and diagnosis. As the development of tracers comes hand-in-hand with the development of potent and selective receptor ligands, there is a rising number of PET tracers available in preclinical and clinical studies. This review analyzes the development of P2X7R positron emission tomography (PET) tracers and their potential in various PET imaging applications.

## 1. Introduction

When Geoffrey Burnstock discovered the involvement of adenosine triphosphate (ATP) in signal transduction via specific receptors in the late 1960s, the initial results were discussed as controversial. Nevertheless, the interest in purinergic signaling grew over the following decades regardless of the initial criticism [1,2,3,4]. The discovered receptors can be broadly subdivided into P1 and P2 receptor families. The P1 receptors, also called adenosine receptors, are G-protein coupled and characteristically activated by adenosine. The P2 receptors, on the other hand, can be further subdivided into two subfamilies: P2X and P2Y receptors. The P2X receptors (P2XRs) are ligand-gated cation channels activated by ATP, while P2Y receptors (P2YRs) belong to the group of G-protein coupled receptors and can be activated by various nucleotides depending on the respective subtype [5].

The P2X receptors assemble as homo- or heterotrimers from the seven known P2X subunits (P2X1-7). The single subunits have their respective N- and C-terminal domains on the intracellular side and are anchored in the cell membrane by two transmembrane domains, which are linked by a large extracellular loop [6]. In 2016, Kawate et al. reported the crystal structure of giant panda (*Ailuropoda melanoleuca*) P2X7R, which revealed the allosteric antagonist-binding pocket located at the intersection of two adjacent subunits juxtaposed to the ATP binding site. Consequently, the trimers bear three allosteric binding sites located in the upper region of the extracellular part [7,8]. More recently, the Cryo-EM structure of the full-length rat P2X7R has also been resolved, while crystallization of human P2X7 has not yet been achieved due to its tendency to aggregate [9].

Amongst the homotrimeric P2X receptors, the P2X7R stands out in its physiological and pharmacological profile. P2X7R activation uniquely requires approximately 10-fold higher concentrations of ATP than other P2X receptor subtypes [10]. Under physiological conditions (transient stimulation), the receptor mediates the rapid influx of Na^+^ and Ca^2+^ ions and the efflux of K^+^ ions together with other cations. Prolonged or repeated activation with ATP (sustained stimulation) stabilizes the formation of larger unselective macropores, which increases the cell wall permeability for molecules up to 900 Da, such as nicotinamide adenine dinucleotide phosphate (NADP). This important functional property, found to a lesser extent in other P2X receptors, is responsible for the cytotoxic activity of P2X7 receptors upon activation/overstimulation [10,11].

The human P2X7R is expressed by virtually all cells of the innate and adaptive immune system. It is found in Schwann cells located in the peripheral nervous system (PNS) [12,13] and in microglia as part of the immune cells of the central nervous system (CNS). In the latter, the highest levels of P2X7R mRNA and protein were found. Besides the immune cells, P2X7R is expressed in a large variety of cell types, including osteoclasts, osteoblasts, Langerhans epidermal cells, and liver cells [14,15]. The receptor is, in lower quantities, also present in neuroglial cells, astrocytes, and oligodendrocytes [16,17,18]. The activation of the P2X7R might play a pivotal role in inflammatory responses leading to the secretion of IL-1beta and IL-18 by activating caspase-1, which promotes the maturation of interleukins and their subsequent release [19,20,21]. As a result, the P2X7R is involved in (neuro)inflammatory processes, and hence is strongly linked to the pathologies of autoimmune diseases such as rheumatoid arthritis (RA), neurodegenerative diseases and neuropathic pain [21,22,23,24]. Most recently, the P2X7R was considered an alternative for the 18 kDa translocator protein (TSPO) as a biomarker for inflammatory processes in the brain and the periphery. The genetic polymorphism of TSPO often leads to a loss of in vitro and in vivo affinity of radioligands and makes the development of specific tracers quite challenging [25,26], therefore creating a need for the evaluation of alternative specific targets such as, for example, P2X7Rs.

Moreover, the overexpression of the P2X7R was found in many kinds of tumors, such as lung, breast, leukemia, neuroblastoma, glioma, cervical, bladder, and bone cancer [27,28,29]. Its role in tumor progression is diverse, having functions related to the development and spread of cancerous cells. Elevated ATP concentrations in the tumor microenvironment (TME) compared to healthy tissue do not lead to apoptosis but rather enhance cancer cell migration [28,30,31]. Possible reasons for this effect are either the expression of P2X7R variants that are unable to form macropores leading to apoptosis or that the activation through raised ATP levels is simply not sufficient due to rapid ATP metabolism via CD39 and CD73 leading to a rise in the concentration of the immunosuppressive and proangiogenic adenosine in TME [32,33,34]. The involvement of P2X7R in cancer progression has led to the increasing interest in P2X7R antagonists for the treatment of cancer [24,35].

The described involvement of P2X7R in various diseases and its expression in cells of the CNS and in tumor cells makes it a promising imaging target to monitor the progression of pathologies related to (neuro)inflammation and cancer. The application of P2X7R positron emission tomography (PET) tracers can visualize the receptor expression and allow in vivo diagnosis and therapeutic monitoring.

### P2X7R Species Differences and Polymorphism

The gene encoding the P2X7R subunit has been reported in at least 55 species. The human P2X7R gene is comprised of 13 exons and is localized in chromosome 12q24.31. [36]. The amino acid sequence of the P2X7R shows high inter- and intraspecies variation. In comparison to the human P2X7 subunit, nonhuman mammalian and non-mammalian subunits show 77–97% and 42–47% sequence identity, respectively [37]. Notably, as two of the most common model organisms in preclinical studies are mouse and rat, the murine subunit has 81% and the rat subunit has 80% sequence identity to the human subtype [38,39]; the mouse and rat subunits share 85% sequence identity [37]. These differences lead to variations in agonist/antagonist potency between the species orthologues, as addressed in various studies that identify receptor regions that contribute to such disparities. In particular, residues at positions 127 and 284 of the P2X7 subunit have been shown to affect agonist potency significantly at rat, human and mouse receptors [37,38]. Additionally, phenylalanine F95 has been identified to allow pi-stacking and, therefore, to be crucial for the binding and affinity of many antagonists at the human receptor. The rat isoform, on the other hand, contains leucine L95 in this position, decreasing the affinity of the ligands and leading to different results between human and rodent studies [40]. A study emphasizing potential differences showed that EC_50_ values for agonists can strongly vary among species. BzBzATP, for example, activates rat and human P2X7Rs at 10 times greater concentrations than mouse P2X7Rs [41].

Being aware of the heterogeneity of the P2X7Rs between species, it becomes obvious that there is a need for more reliable models. In cancer-related studies, xenograft models have become increasingly popular over the years. In these models, tumor samples expressing the human P2X7R can grow in physiologically-relevant tumor microenvironments mimicking the oxygen, nutrient, and hormone levels that are comparable to the human environment [42]. Nevertheless, the primary grafts in mice lack heterogeneity due to the homogeneity of the implanted cell line [43]. This issue can be overcome by implanting tumor tissue directly from the patient into the model animal (mostly rats and mice), known as patient-derived xenografts (PDX). The implanted tumor tissue maintains the desired genetic and epigenetic heterogeneity found in the patient [44]. As a result, PDX models have been used more frequently in preclinical studies of P2X7R antagonists [45,46]. However, it must be mentioned that immune-deficient mice strains must be used to prevent graft rejection [47]. This could potentially compromise studies of compounds targeting parts of the immune system. Regardless of the source of the xenograft, the host (mouse or rat) expresses the species-specific P2X7R form in its tissue with a potentially different activity profile towards the tracer, thereby altering the possible background signal in imaging applications, which must be considered in tracer development and evaluation.

Another important aspect to be taken into account in P2X7R tracer development is the P2X7R diversity caused by the approximately 150 non-synonymous single nucleotide polymorphisms (SNPs), many of which have not been intensively investigated to date [48]. A total of 16 SNPs coding for missense mutations have been characterized in the coding region of the human P2RX7 gene [49]. Various SNPs alter P2X7R activity, with many either increasing (gain-of-function) or decreasing (loss-of-function) susceptibility towards agonists and antagonists [50]. In particular, the P2X7B and nfP2X7R isoforms lacking the macropore-forming cytotoxic activity appear to be of great importance for cancer progression due to their protumorigenic activities and are discussed as potential prognostic biomarkers [51]. However, a comprehensive analysis of the expression of P2XR isoforms in various types of cancer is still not available.

Due to the described variety of P2X7Rs, it can be assumed that experimental results could differ largely from each other and are highly context-dependent, especially between different model organisms, making the translation from animal studies problematic. In practice, the decorrelation between preclinical and clinical results is commonly observed [52,53,54].

P2X7R positron emission tomography (PET) tracers can visualize the receptor expression and allow in vivo diagnosis and therapeutic monitoring. There are already various candidates in diagnostic applications at different stages of preclinical and clinical evaluation [55,56,57]. The following section will illustrate the P2X7R antagonists in PET imaging applications, their ability to penetrate into the CNS, or their bioavailability in the periphery, and their translation from preclinical into clinical imaging applications.

## 2. P2X7 Receptor Imaging Tracers

P2X7R antagonists can be broadly subdivided into two categories: those able to penetrate the blood-brain barrier (BBB) and enter the central nervous system, or those remaining peripherally. Commonly linked CNS P2X7R applications are diseases like Alzheimer’s disease (AD), Parkinson’s disease (PD) or multiple sclerosis (MS) [58], as well as the formation of different types of cancer, i.e., glioblastoma multiforme (GBM) [59]. On the other hand, peripherally bioavailable P2X7R antagonists that are not BBB-permeable are attractive candidates for the treatment/diagnosis of lung and breast cancer [60,61].

Due to the rising interest in the P2X7R, the development of several compound classes through academia and the involvement of many pharmaceutical companies are ongoing. This is also displayed in the registration of patents on a range of structurally diverse P2X7R antagonists (Figure 1) [62,63].

### 2.1. Radiolabelling Strategies

The synthesis of radiolabeled P2X7R ligands comes with certain requirements. Due to the radioisotopes’ short half-life time, fast and efficient labeling methods that diverge from the classical pathways in organic synthesis are necessary. Two of the most common isotopes applied in PET imaging of P2X7R expression in vivo are carbon-11 and fluor-18.

The primary source of [^11^C]carbon is the ^14^N(p,α)^11^C reaction on nitrogen gas [64], doped with oxygen or hydrogen, leading to one of the two primary labeling agents, [^11^C]carbon dioxide and [^11^C]methan. The former in particular can be subject to carrier dilution from atmospheric carbon dioxide, while methane lacks sufficient reactivity. [^11^C]carbon dioxide allows the introduction of carboxyl groups, but the resulting carboxylate-bearing ligands often display limited blood-brain barrier (BBB) permeability. Therefore, the primary labeling agents are commonly converted further to enable fast and efficient reactions with precursor molecules and diversify the accessible structures. One of the most widely applied secondary labeling agents is [^11^C]methyl iodide, which can be obtained from both [^11^C]carbon dioxide as well as from [^11^C]methan. The former is reduced to ^11^CH_4_ and subsequently substituted with HI or by reaction with I_2_ under high pressure [65]. The ‘on-line’ conversion of [^11^C]methyl iodide to [^11^C]methyl triflate can yield another more reactive ^11^C-methylating agent [66,67]. A practical example for the incorporation of [^11^C]carbon into a P2X7R ligand is the *N*-methylation of [^11^C]GSK1482160 utilizing the activated secondary labeling agent [^11^C]CH_3_OTf, which was obtained by the nucleophilic substitution of [^11^C]CH_3_I with silver triflate and then was used to methylate the corresponding amide (Figure 1). The tracer was obtained in high specific activity 260–360 GBq/μmol and radiochemical yield (rcy) of 30–40% [68].

For the generation of [^18^F]fluoride, the ^18^O(p,n)^18^F reaction on ^18^O-enriched water has become almost universally applied [69]. The yielded [^18^F]fluoride ion can primarily be used in aliphatic nucleophilic substitution reactions, aromatic nucleophilic substitutions, and transition metal-catalyzed fluorination reactions. The [^18^F]-JNJ-64413739 was synthesized via a nucleophilic substitution reaction of [^18^F]fluoride in the presence of kryptofix, potassium oxalate (K_2_C_2_O_4_) in deuterated dimethyl sulfoxide (Figure 1). The PET tracer was prepared with a specific activity of 17–100 GBq/μmol. However, the authors did not report on the radiochemical yields (rcy) of the reaction.

**Scheme 1 ijms-24-01374-sch001:**
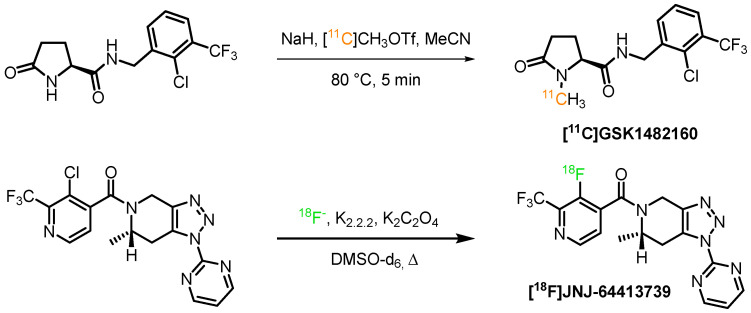
Examples of radiosyntheses of [^11^C]GSK1482160 [^18^F]-JNJ-64413739 [68,70].

### 2.2. P2X7 Receptor Tracers for the PET Imaging in CNS

The CNS consists of the brain and the spinal cortex and is arguably the most complex organ in our body. It has to adapt to changing environmental challenges, and to achieve this, it is isolated from the rest of the organism by the blood-brain barrier so that it can control and protect itself efficiently. The BBB enables the selective uptake of molecules as a protection mechanism. Consequently, brain-specific diseases like the aforementioned AD or PD can only be monitored and treated by compounds that cross the BBB.

The aforementioned abundance of P2X7 In the CNS and its involvement in various diseases sparked excitement in the scientific community, resulting in the advancements of tracers for imaging of neuroinflammatory responses in CNS pathologies (Table 1).

GSK1482160 (Table 1), a pyroglutamic acid amide analogue produced by GlaxoSmithKline (GSK), demonstrated high potency in in vitro assays on rat and human P2X7Rs, together with an excellent safety profile and potency in in vivo rat models [92]. These initially promising results led to the subsequent investigation of its pharmacokinetic (PK) and pharmacodynamic (PD), safety and tolerability parameters in healthy human subjects [57]. The study revealed a desirable pharmacological profile; however, it was not possible to achieve sufficient P2X7R inhibition whilst maintaining a necessary safety margin with regard to the applied dose. These findings led to a temporary halt on the further development of GSK1482160 to a therapeutic for chronic inflammatory pain [57]. Interestingly, it was discovered that GSK1482160 also showed an ability to cross the BBB and enter the CNS, making it potentially useful in a neurological context [56]. This shifted the focus to the development of a PET tracer as a potential biomarker of neuroinflammation, reviving the interest in the structure. Recent publications about GSK1482160 and derivatives exclusively contain the development of radiolabeled [^11^C]GSK1482160. The radiolabeled compound can bind to the P2X7R and indicate the extent of expression and progression of neuroinflammation [74]. Green et al. concluded from a clinical trial (NCT00849134) in healthy subjects that even though the brain uptake was as low as 2% of the injected dose, the compound appears suitable for PET studies of the P2X7R expression for monitoring inflammatory processes [71]. In a different approach, Gao et al. tried to address the problem of low brain uptake by exchanging the aromatic chlorine for the homologous halogen substituents, forming the series [^11^C]halo-GSK1482160 (F-, Br-, and I-); with the result that the bromo- and iodine-bearing compounds were binding with a slightly higher affinity to P2X7R in an in vitro competitive binding assay on human P2X7R (HEK293-hP2X7R) [76]. In vivo studies still need to be conducted in order to confirm these initially promising results and show whether the addressed issues can be solved to obtain a promising candidate for clinical trials. In order to extend the available time frame for efficient synthesis and imaging studies, the GSK1482160 lead structure was modified by the introduction of the fluoroethyl substituent at the lactam portion to yield an equally P2X7R potent, 18-fluorine labeled PET tracer [^18^F]IUR-1601. Careful optimization of the radiolabeling reaction conditions was required to suppress the competing elimination reaction leading to the formation of the vinyl side product and yielding [^18^F]IUR-1601 in moderate molar activities and radiochemical yields [75]. The results of in vivo imaging applications using [^18^F]IUR-1601 have yet to be reported.

Janssen et al., developed adamantane benzamide-based tracers, of which [^11^C]-SMW139 and [^11^C]SMW64-D16 underwent further evaluation [78]. [^11^C]SMW64-D16 targeted inflamed areas in brain slices of two rat models but did not show sufficient brain uptake in rodent biodistribution studies via PET [93]. The trifluorinated variant of the adamantane benzamide [^11^C]SMW139, did not show a difference in tracer binding affinity between the tissues of AD patients and healthy subjects ex vivo. However, there were promising results in an in vivo experimental autoimmune encephalomyelitis (EAE) rat model with regard to the tracer uptake at the peak of the disease in neuroinflammation imaging with the P2X7R PET tracer [^11^C]SMW139 [94]. Most recently, with the first human studies (NCT04126772), the compound entered clinical trials, although only with a small number of subjects. It demonstrated good pharmacokinetic properties and brain uptake; moreover, higher signals were obtained from 90-min dynamic PET scans in active relapsing-remitting multiple sclerosis (RRMS) patients compared to the control group of healthy subjects [95]. As a next step, larger cohort studies, controlling for varying demographic and lifestyle factors, would undeniably be illuminating.

In a similar fashion, another tracer, [^11^C]JNJ-54173717, was tested in a rat model with local overexpression of the human P2X7R (hP2X7R), realized by injecting the animal with a viral vector carrying the human receptor gene. The in vitro study on vector-injected rats’ transversal rat brain sections showed higher tracer binding than in the negative control. The same study could also demonstrate the tracer’s ability to cross the BBB as well as the selectivity of [^11^C]JNJ-54173717 towards the P2X7R in nonhuman primates [84]. In two well-characterized rat models of PD, using in vitro autoradiography at multiple time points throughout the disease progression, a time-dependent increase in tracer binding was observed in one of the models, where the brains of rats were treated with 6-hydroxydopamine (6-OHDA). However, in a chronic A53T viral vector model of PD, no such correlation was observed [81]. From a first-in-human study (NCT03088644) with 11 healthy subjects and 10 PD patients, Van Weehaeghe et al. concluded the tracer to be a promising PET radioligand for quantifying P2X7R expression with sufficient brain uptake. They were also able to detect a potential genotypical polymorphism (rs3751143) responsible for different binding affinities. A comparison of PET signal intensity between healthy subjects and PD patients, however, did not reveal a measurable distinction in tracer uptake [80]. A possible explanation would be the time dependence of P2X7R-expression during the progression of the disease, which is more dominant in the early stages. Despite this drawback, [^11^C]JNJ-54173717 may still prove its potential in detecting neurodegenerative and inflammatory diseases in future studies [96].

[^18^F]JNJ-64413739, a selective P2X7R antagonist with an affinity and potency in a low nanomolar range, was developed by Janssen Pharmaceuticals and preclinically tested in a rat lipopolysaccharide (LPS) local neuroinflammation model. For that purpose, rats were intrathecally injected either with LPS in one hemisphere to induce neuroinflammation in brain tissue or injected with phosphate-buffered saline (PBS) as a negative control. The neuroinflammation process was confirmed by the upregulation of three biomarker genes: TSPO, AIF-1, and P2X7R, evaluated by qPCR. Compared to the negative control, the expression level of P2X7R was elevated by factor two and proved this receptor once more to be a promising target for the imaging and treatment of neuroinflammation. In the preclinically performed PET imaging studies, a significant increase of the PET signal in LPS-injected rats was observed relative to both the PBS-injected control and the naïve animals. The specificity of tracer uptake in the LPS-injected hemisphere was demonstrated by blocking studies with JNJ-54175446 as an inhibitor. Compared to vehicle-control animals, the pre-treatment with JNJ-54175446 10 min before starting the PET imaging lowered the tracer uptake in the LPS-injected as well as in the non-injected hemisphere significantly., The decrease of tracer uptake was higher than at the control site supporting the hypothesis that the [^18^F]JNJ-64413739 tracer signal is specific to the P2X7R expression mainly in the LPS-injected hemisphere [97]. Further preclinical studies from Kolb et al. investigated the level of P2X7R occupancy by the [^18^F]JNJ-64413739 tracer in healthy rodents. Here, the expression and activity levels of the P2X7R under these physiological conditions were expected to be low. Nevertheless, the P2X7R occupancy study in healthy rats demonstrated the specificity of the [^18^F]JNJ-64413739 tracer by using JNJ-55308942 as an inhibitor. The one hour pre-treatment with the inhibitor prior to PET imaging resulted in a decrease in tracer uptake in the brain under physiological conditions. An additional microdosing approach with the [^18^F]JNJ-64413739 tracer using a P2X7R knock-out (KO) mouse model suggested a certain degree of non-specific binding of the tracer in the brain tissue. The non-specific binding was also observed in a P2X7R occupancy study in healthy adult rhesus macaques (*Macaca mulatta*) as a nonhuman primate organism. Matching with previous data, the [^18^F]JNJ-64413739 tracer accumulates in the brain tissue under physiological conditions and could be reduced in a dose-dependent manner by pre-treatment with JNJ-54175446 as an inhibitor. Furthermore, in these experiments, a certain degree of non-specific binding of the [^18^F]JNJ-64413739 tracer could be observed to be in agreement with the previous data from the rodent experiments [86]. The high affinity, specificity, metabolic stability, and low protein-bound fraction in plasma make the [^18^F]JNJ-64413739 tracer suitable as a clinical imaging agent. Koole et.al. evaluated the clinical biodistribution, the pharmacokinetic profile, and the P2X7R occupancy in 16 healthy subjects (NCT03088644, NCT03437590). The presented data of [^18^F]JNJ-64413739 supports its assessment as a reliable PET tracer for quantification of P2X7R in the brain tissue to study its involvement in neuroinflammation, neurodegeneration, and mood disorders or to evaluate the occupancy level of selective BBB permeating P2X7R inhibitors [70]. Most recently, a structurally related P2X7R targeting PET tracer [^18^F]FTTM was successfully applied in the rat model of temporal lobe epilepsy. Tracer uptake was associated with activated microglia and proved the potential of P2X7R imaging for monitoring neuroinflammation again [87].

The attempted subtle structural modifications in GSK1482160 and JNJ64413739 derived series of PET tracers are rather surprising as they have led to significant differences in the performance of the respective PET tracer, which leads to the question of the predictivity of the available in vitro models for assessing the pharmacokinetic profiles in tracer development.

The first P2X7R imaging ligand that went into in vivo testing was [^11^C]A740003, developed by Abbott Laboratories, after showing promising results in initial in vitro assays. However, the in vivo studies showed a low brain uptake and a moderate metabolic rate in mice, disqualifying [^11^C]A740003 for the efficient imaging of the brain [91]. In an approach to overcome the uptake issues, [^18^F]EFB, another tracer bearing a cyanoguanidine structure, which derived from A804598, was developed. A804598 is known to be a blood-brain barrier permeable P2X7R antagonist, so the structural resemblance appeared to be promising [98]. In vitro testing in a calcium influx binding assay revealed low nM binding constants for the human P2X7R, but a 200-fold higher binding constant for the murine isoform [79]. The compound displayed a low brain uptake in rats, showing potentially limited suitability of the cyanoguanidine moiety for BBB uptake. The ligand’s high potency for the human receptor and low activity at the murine/rat subtype suggests a (misleadingly) low background for potential evaluation in xenograft models of human tumor tissues implanted into mice.

There is a growing number of imaging ligands for applications in the CNS targeting the P2X7R due to its vast distribution in glial cells and, consequently, its involvement in neuroinflammatory processes (due to P2X7R’s mediatory role in the release of IL-1β and the activation of the NLRP3 inflammasome). The majority of ligands discussed in this review showed good results in preclinical in vitro and in vivo evaluations, and some have already proceeded to clinical trials, albeit with a limited number of participants. With regard to these trials, it becomes clear that the in vivo brain uptake in human patients remains a challenging task for current imaging tracers. Another difficulty for the translation of in vitro into in vivo studies involves the heterogeneity in the disease progression, e.g., the time dependence of P2X7R-expression and the expression of different P2X7R variants in the tissue. The specific imaging of acute neuroinflammatory diseases in comparison to healthy individuals displayed an additional challenging task for some of the tested tracers, as with [^11^C]JNJ-54173717. Neither [^11^C]A740003 nor [^18^F]EFB showed appreciable brain uptake; nevertheless, data exist which suggest the use of those tracers outside the CNS, which shall be discussed in the following section.

### 2.3. P2X7 Receptor Tracers for the Imaging Applications Outside of CNS

Of widespread interest is the role of P2X7R antagonists not only in the CNS but also in the periphery. This includes inflammatory processes, as well as different types of cancer. Tracers showing limited BBB permeation but high P2X7R binding in vivo might provide promising results in peripheral imaging applications.

In 2019, Fu et al. revealed a novel P2X7R imaging ligand, [^18^F]PTTP, which might be useful in the screening of new drugs and in distinguishing inflammation from lung tumors [89]. [^18^F]PTTP is based on a triazolo-tetrahydropyridine structure related to [^18^F]JNJ-64413739 developed by Janssen Pharmaceuticals. PTTP showed promising physicochemical properties, excellent pharmacokinetic characteristics, and a strong affinity to hP2X7 and mP2X7 receptors, and was therefore selected for inflammation imaging. Compared to the aforementioned JNJ compounds, [^18^F]PTTP showed a low brain uptake, which is, however, higher than for [^11^C]A740003 and [^18^F]EFB [84,91]. From their initial results, Fu et al. concluded that [^18^F]PTTP is a better probe to localize inflammatory diseases by screening for macrophages than to identify lung tumors due to the low tumor uptake in xenograft models of human lung cancer cells in mice. However, the study addresses the great potential of [^18^F]PTTP as a PET tracer and P2X7R antagonist for screening new drugs, quantifying the expression of P2X7R in peripheral inflammation, and distinguishing inflammation from the tumor [89]. A structurally related PET tracer [^18^F]FTTM was not only demonstrated to provide sufficient brain uptake for the imaging of neuroinflammation in the rat model of temporal lobe epilepsy, but also to outperform [^18^F]FDG in the imaging of the ApoE Mouse model of atherosclerotic plaques [87].

Finally, there are numerous ligands that were proven to modulate the P2X7R activity in the search for new therapeutic strategies, such as AZ10606120 [99,100], A-438079 [101], AZD9056 (NCT00908934, NCT00920608, NCT00700986, NCT00736606, NCT00520572), CE-224535 (NCT00628095), and GSK1482160 (NCT00849134). Subtle structural modifications to introduce radioactive isotopes are rather unlikely to impact the affinity of the ligand to the receptor significantly; consequently, already known ligands with high affinity and selectivity for the P2X7R bear the potential to function as a starting point for the development of novel radiotracers.

## 3. Conclusions

In search of new strategies to diagnose neuroinflammation and cancer, the P2X7R is increasingly gaining attention as a promising new target. The species differences in P2X7R structure appear not to pose a hurdle for translatability from preclinical model organisms to mature clinical applications since various P2X7R targeting PET tracers have successfully entered clinical imaging studies. However, the P2X7R polymorphism might still pose an additional challenge for transition into the clinic, especially when investigating P2X7R expression in cancer. Here more studies evaluating the predominant P2X7R form in different types of cancer would be of great importance.

Subtle structural modifications, as observed for GSK1482160- or JNJ64413739-derived PET tracers, strongly affected the outcome of the imaging studies, in particular when dealing with CNS bioavailability. Here the integration of reliable in vitro models to assess the pharmacokinetic parameters would accelerate the selection, optimization, and development of potent P2X7R targeting PET tracers.

Despite these difficulties, the race for imaging probes capable of providing the requisite tools to elucidate the P2X7R expression and its role in diseases related to (neuro)inflammation and cancer is on. The P2X7R is a bright new target with great potential to be broadly explored in the near future.

## Data Availability

Not applicable.

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
