# Peer review of "Spotlight on P2X7 Receptor PET Imaging: A Bright Target or a Failing Star?"

_ijms, 2023, doi:10.3390/ijms24021374_

Round 1

Reviewer 1 Report

In the presented manuscript Authors provide a deep and comprehensive discussion on the application of P2X7 receptor tracers in diagnosis of inflammatory diseases and cancers. They broadly discuss the biological background of target receptor and its role in relevant diseases. In the section devoted to the recently developed tracer compounds, they provide an extensive summary of the current achievements in the field.

The manuscript is written in good English, with the well-organized structure. I’ve found only a few stylistic errors and typos that could be easily corrected. They are listed below:

line 22: “discussed controversially” – change to “discussed as controversial”

line 53: immune system of the central nervous system – if possible, avoid using “system” twice

line 84: “intraspecific” – change to “intraspecies”

line 319: “high protein-binding free fraction” – would sound better as: “low protein-bound fraction”

As for the substantial content of the manuscript, I would have two comments:

1.       In the introduction (lines 67-76), Authors discuss the role of ATP in apoptosis of cancer cells, and suggest that lack of ATP activity in promoting cell death is due entirely to insufficient activation of P2X7. However, it is known that other P2 receptors are expressed in cancer environment and that ATP metabolites like adenosine can also act through relevant receptors expressed in the tumor site. This aspect should also be mentioned in the discussion.

2.       As discussed by the Authors, for many of P2X7R tracers there is a discrepancy between good receptor labelling properties and satisfactory penetration to CNS on one hand, and lack of significant differences in PET imaging of healthy and pathological tissues on the other side. Authors discussed this inconsistency e.g. in context of JNJ-54173717. Could Authors provide a broader discussion on the reasons for failures of diagnostic use of P2X7R tracers rising from P2X7R biology and disease state themselves, rather than from compounds properties?

To summarize, the reviewed manuscript provides a broad overview of a hot scientific topic and is of high value for the community. I highly recommend the presented manuscript for publication after addressing the minor points, I have mentioned above.

Author Response

We wish to thank the careful review of this manuscript by the referees; these comments have contributed to the improvement of the manuscript and several suggestions made are relevant and of interest. We have revised the manuscript according to all suggestions.

Reviewer 1:

Comments and Suggestions for Authors

In the presented manuscript Authors provide a deep and comprehensive discussion on the application of P2X7 receptor tracers in diagnosis of inflammatory diseases and cancers. They broadly discuss the biological background of target receptor and its role in relevant diseases. In the section devoted to the recently developed tracer compounds, they provide an extensive summary of the current achievements in the field.

The manuscript is written in good English, with the well-organized structure. I’ve found only a few stylistic errors and typos that could be easily corrected. They are listed below:

line 22: “discussed controversially” – change to “discussed as controversial”

corrected

line 53: immune system of the central nervous system – if possible, avoid using “system” twice

thank you for pointing it out, we have corrected it.

line 84: “intraspecific” – change to “intraspecies”

corrected

line 319: “high protein-binding free fraction” – would sound better as: “low protein-bound fraction”

thank you for pointing it out, we have corrected it.

As for the substantial content of the manuscript, I would have two comments:

  1. In the introduction (lines 67-76), Authors discuss the role of ATP in apoptosis of cancer cells, and suggest that lack of ATP activity in promoting cell death is due entirely to insufficient activation of P2X7. However, it is known that other P2 receptors are expressed in cancer environment and that ATP metabolites like adenosine can also act through relevant receptors expressed in the tumor site. This aspect should also be mentioned in the discussion.

We have added it.

  1. As discussed by the Authors, for many of P2X7R tracers there is a discrepancy between good receptor labelling properties and satisfactory penetration to CNS on one hand, and lack of significant differences in PET imaging of healthy and pathological tissues on the other side. Authors discussed this inconsistency e.g. in context of JNJ-54173717. Could Authors provide a broader discussion on the reasons for failures of diagnostic use of P2X7R tracers rising from P2X7R biology and disease state themselves, rather than from compounds properties?

 We have added some additional discussion; however, depending on the condition/disease, there are many aspects to be taken into account which would go beyond the scope of this review.

To summarize, the reviewed manuscript provides a broad overview of a hot scientific topic and is of high value for the community. I highly recommend the presented manuscript for publication after addressing the minor points, I have mentioned above.

Reviewer 2 Report

This is an interesting and nicely written review focusing on the development of P2X7 Receptor ligand positron emission tomography (PET) tracers’ identification and clinical application. After several introductive sections on P2X receptors and P2X7R gene/polymorphisms, the authors explored in detail the possible application of P2X7R Imaging Tracers for the elucidation of the P2X7R expression and its role in diseases such as cancer and inflammation. Several radiolabeled receptors’ agonists and related pre-clinical in vivo/in vitro and clinical studies, as well as agonists’ clinical applications are described in detail. In my opinion the manuscript is well written and matches adequately with the aim and scope of IJMS. Please see below several minor observations for improving the work.
1.    Line 19 “Introduction” should be aligned.
2.    Line 8 I would also include inflammation, as reported doi: 10.1016/j.immuni.2017.06.020.
3.    PET should be positron emission tomography when first mentioned
4.    Line 51 Besides immune cells, P2X7R is expressed in a large variety of cell types, including osteoclasts and osteoblasts, Langerhans epidermal cells, liver cells. For completeness, this information should be included.
5.    For completeness, I would include a couple of words on the other members of P2X family (P2X1-7) in the introductive section.
6.    Section 1.1 The P2X7R gene structure should be, at least briefly, described, alongside the most important loss-of-function human SNPs which have been currently identified
7.    Line 83 This 2022 review on P2X7 receptor on Cancers MDPI journal should be included (DOI: 10.3390/cancers14051116)
8.    Please include the complete name for EC50 for non- expert readers
9.    Lines 161-166 These sentences seem to be redundant and would fit better at the end of the introduction (line 140)
10.    Line 171 in human or animal models?
11.    Authors should carefully evaluate the writing style of different scientific words and uniform their mentioning. For instance line 231 “P2X7R” and line 234 “P2X7 receptor”
12.    Line 232 (but also other sections of the review) If clinical trials are mentioned, the ID from www.clinicaltrial.gov should be included.
13.    Line 366 cancers’ names?
14.    Lines385-390 other P2X7R ligands have been theorized/identified in clinical trials as antitumor agents such as A740003, AZD9056 and CE-224535
15.    Line 408 “ro)inflammation”
